# Three-Station Non-Contrast MR Angiography of the Lower Extremities Using Standard and Centric Fresh Blood Imaging

**DOI:** 10.3390/s25247429

**Published:** 2025-12-06

**Authors:** Won C. Bae, Anya Mesa, Vadim Malis, Yoshiki Kuwatsuru, Katsumi Nakamura, Ann Gaffey, Mitsue Miyazaki

**Affiliations:** 1Department of Radiology, University of California-San Diego, La Jolla, CA 92093, USA; wbae@health.ucsd.edu (W.C.B.); anmesa@health.ucsd.edu (A.M.); v2malis@health.ucsd.edu (V.M.); 2Department of Radiology, Juntendo University School of Medicine, Tokyo 113-8421, Japan; ykuwatsuru@health.ucsd.edu; 3Tobata Kyoritsu Hospital, Kitakyushu 804-0093, Japan; nakamura.katsumi@gmail.com; 4Department of Surgery, University of California-San Diego, La Jolla, CA 92037, USA; agaffey@health.ucsd.edu

**Keywords:** non-contrast MR angiography, fresh blood imaging (FBI), peripheral artery disease, time reduction, image quality

## Abstract

**Background**: Peripheral artery disease (PAD) is a manifestation of atherosclerosis that affects the extremities, leading to reduced perfusion and functional impairment. Non-contrast magnetic resonance angiography (NC-MRA) provides a safe and quantitative approach for early detection of PAD without the risks associated with contrast agents. The purpose of this study was to demonstrate the application of standard and centric *ky*-*kz* FBI techniques for rapid three-station NC-MRA of the entire lower extremity. **Methods**: This prospective cross-sectional study compared standard three-station fresh blood imaging (sFBI) with centric *ky*-*kz* ordered fresh blood imaging (cFBI) sequences in 10 healthy subjects and 3 patients with PAD (age range: 23–79 years; 7 females) using a 3-Tesla magnetic resonance imaging (MRI) system. Both sequences were acquired at the iliac, femoral, and tibial stations. Image quality (0–4 scale), signal-to-noise ratio (SNR), and contrast-to-noise ratio (CNR) were evaluated. Statistical analysis was performed using repeated-measures analysis of variance (ANOVA) with significance set at α = 0.05. **Results**: Image quality did not differ significantly between sFBI and cFBI (*p* = 1.0). The iliac station exhibited lower image quality than the femoral station (*p* < 0.01). In a PAD patient with an iliac stent, cFBI preserved good image quality in the femoral and tibial stations, whereas sFBI was affected by N/2 aliasing artifacts. Both methods failed to visualize the stented iliac segment. Compared to sFBI, cFBI yielded significantly lower SNR (*p* < 0.01) and CNR (*p* < 0.001) but reduced total scan time by approximately 40% (468 s vs. 291 s). **Conclusions**: Three-station non-contrast FBI MRA of the peripheral arteries is feasible. The cFBI sequence substantially shortens scan time without compromising diagnostic image quality, offering practical advantages for clinical implementation, improved patient comfort, and reduced motion artifacts.

## 1. Introduction

Peripheral artery disease (PAD) results in atherosclerosis of the extremities and affects more than 10 million people in the United States [1]. It results in the narrowing or blockage of blood vessels due to plaque accumulation. Detecting PAD early is important, as it can help prevent serious complications [1] such as infections, limb amputation, and even death. Although there are various imaging techniques available to diagnose PAD, each has its drawbacks. Intra-arterial digital subtraction angiography is considered the gold standard [2], its use has declined for pure diagnostic purposes due to its invasive nature, and it is often used during a procedure. Computed tomography angiography (CTA) is widely used for its high-resolution images, but it involves exposure to radiation and the use of contrast agents—factors that pose risks for patients [3,4,5], especially those with diabetes and kidney issues [6]. Contrast-enhanced magnetic resonance angiography (CE-MRA) [7,8,9] offers excellent image clarity, but it relies on gadolinium-based contrast agents (GBCAs), which have been linked to conditions like nephrogenic systemic fibrosis (NSF) [10] and long-term accumulation in the brain and other tissues [6,11,12,13]. Additionally, many PAD patients with poor kidney function are not able to receive GBCAs. In contrast, non-contrast MRA (NC-MRA) [14,15,16,17,18,19,20] avoids these concerns entirely.

NC-MRA has emerged as a compelling alternative to CE-MRA, particularly for patients with impaired renal function. A series of foundational review articles [21,22,23] have established the clinical importance, physical principles, and recent technical advancements of NC-MRA, including techniques such as time-of-flight (TOF) [24], quiescent-interval single-shot (QISS) [16], phase contrast (PC) [25], and fresh blood imaging (FBI) [14,15,26]. The TOF technique, developed for brain arterial imaging, is unsuitable for the lower extremities due to the long scan time [27], slower blood flow in the peripheral vasculature [28,29], and saturation artifacts [28]. Phase contrast MRI is also generally not suited, due to long scan time and image artifacts from turbulent and slow flows [28,30]. FBI (Figure 1) has gained traction due to its reliance on intrinsic physiological blood flow differences across the cardiac cycle. Standard FBI (sFBI; Figure 1B, left), introduced through early technical developments [14,15], was optimized for peripheral vascular imaging by incorporating ECG gating and flow-spoiled gradients to separate arterial and venous signals. Clinical validation at 1.5T demonstrated high sensitivity and specificity for peripheral arterial disease screening [26]. Besides non-contrast MRA, sFBI was also utilized in non-contrast MR venography for detecting deep vein thrombus [31,32]. More recently, centric FBI (cFBI; Figure 1B, right) has been proposed as a faster alternative, using zigzag centric *ky*-*kz* k-space trajectories and exponential refocusing flip angles to restore longitudinal magnetization and reduce scan times while preserving image quality [33]. Comparative studies have shown that cFBI significantly reduces scan time but increases T2-related blurring in the slice direction relative to sFBI; however, it can be particularly reduced when combined with parallel imaging techniques [34]. Moreover, zigzag centric *ky*-*kz* trajectory with a radar-like filling reduces motion artifacts in cFBI with 3D single-shot FSE (SSFSE) readout [33] and fast gradient echo readout (cFGE) in MR coronary angiography [35].

Three-station magnetic resonance angiography (MRA) is a well-established imaging protocol for evaluating PAD, enabling comprehensive visualization of the lower extremity arterial tree by dividing it into pelvic, thigh, and tibial stations. Each station captures a different segment of the vasculature, for the evaluation of the entire tree. This multi-station approach is essential for detecting multi-level atherosclerotic disease and assessing the extent of stenosis, occlusion, and collateral flow, which are critical for treatment planning. However, the extensive anatomic coverage required, often 1 m or longer from the renal arteries to the ankles, presents many challenges. Particularly for contrast-enhanced MRA, either multiple injections are needed, or the timing of bolus-chasing must be precise [36]. For NC-MRA techniques, challenges include long scan times (~7 min [37] using QISS [16]), sensitivity to the patient’s blood flow, and low signal-to-noise ratio [21,22]. Recent advancements in NC-MRA, including centric *ky*-*kz* FBI, have the potential to improve the clinical feasibility of three-station MRA of the lower extremity by reducing scan time without compromising imaging quality and diagnostic accuracy [26]. Three stations are required because the iliac, femoral, and tibial arteries represent distinct imaging challenges, and sFBI and cFBI likely perform differently depending on vessel size, depth, and flow. Using all three stations ensures a valid, clinically representative comparison.

The purpose of this study was to demonstrate the application of standard and centric *ky*-*kz* FBI techniques for rapid three-station NC-MRA of the entire lower extremity. We compared the standard (sFBI) vs. centric *ky*-*kz* FBI (cFBI) acquisitions at each station on the scan time, image quality score, signal-to-noise ratio (SNR), and contrast-to-noise ratio (CNR) in healthy human subjects as well as patients with PAD.

## 2. Materials and Methods

This human subject study was approved by the institutional review board. All procedures performed in this study involving human participants were compliant with the regulations of the Health Insurance Portability and Accountability Act (HIPAA), and informed consent was obtained.

**Human Subjects:** Healthy volunteers (*n* = 10, 5 females, age range = 23 to 79 years old, mean age of 47.3 years old with a standard deviation of 17.1 years old) along with patients with PAD (*n* = 3, 46- and 53-years old males, and a 74-years old female) were recruited for this study (7 males and 6 females total). Inclusion criteria were ages between 18 and 80 years and patients either being healthy (no current pain or symptoms related to PAD) or having no known vascular conditions of PAD. Exclusion criteria included counter-indications for receiving MRI (such as claustrophobia, metal in the body, etc.). The age, sex, and resting heart rate of each subject are listed in Table 1.

**Standard FBI vs. Zigzag Centric *ky*-*kz* FBI:** While sFBI has shown a high sensitivity and specificity for stenosis detection [26], it has intrinsic drawbacks of motion artifacts due to long scan time and Nyquist (N/2) artifacts. The newly introduced cFBI addresses these issues [33]. The key distinction between sFBI and cFBI is in their k-space traversal strategies and their impact on flow-related artifact robustness and scan efficiency. sFBI acquires k-space using conventional Cartesian *ky*-*kz* ordering [15], which distributes flow-related phase errors and motion artifacts across the entire acquisition window. N/2 artifacts in the phase encoding (PE) direction are often observed due to inappropriate read-out (RO) dephasing spoiler pulses. This may lead to degraded subtraction image quality in regions of high or turbulent flow and motion. In contrast, cFBI utilizes a zigzag centric *ky*-*kz* trajectory that prioritizes central k-space early after cardiac gating, capturing the most contrast-relevant data before substantial flow or motion artifacts accumulate [33]. This results in improved subtraction efficiency and vessel conspicuity [33]. Moreover, cFBI integrates asymmetric Fourier imaging (AFI) and omits the corners of k-space, which not only shortens overall scan time but also reduces the sensitivity to motion and N/2 artifact propagation.

**MR Imaging:** Bilateral lower extremities of the participants were imaged at 3-T (Galan, Canon Medical Systems Corp., Otawara, Japan) using two 16-channel body array coils covering the entire legs and the pelvis, in conjunction with a posterior 18-channel spine coil. Two FBI sequences (sFBI and cFBI) were used to image the three stations encompassing the iliac, femoral, and tibial arteries. Both used single-shot fast spin echo (SSFSE) acquisition with read-out in the superior–inferior direction, repetition time (TR) = 2 R-R intervals or 1196 to 2804 ms (dependent on the heart rate, HR), echo time (TE) = 60 ms, field of view (FOV) = 450 (phase encode, PE) × 400 (readout, RO) mm, matrix PE × RO = 320 × 288, flip angle/refocusing flip angles = 90/180 deg, 60 to 90 slices, 2.6 to 3.0 mm slice thickness, spectral attenuated inversion recovery (SPAIR) fat suppression, 1D parallel imaging factor of 5 (phase encode or PE direction), and number of averages = 1 for both systolic and diastolic acquisitions. Echo train length (ETL) varied slightly by station: iliac, femoral, and tibial stations had typical ETL values of 60 to 68, 52 to 60, and 40 to 48 for both sFBI and cFBI, respectively. (ETL is slightly longer for cFBI due to the zigzag centric *ky*-*kz* trajectory.) Scan time varied from subject to subject due to differences in HR (Table 1). For each subject and the scan, the actual scan time was recorded to determine the mean and standard deviation (SD) in scan time between the two sequences and three stations. For systolic and diastolic triggering delays, precalculated triggering delays from the HR were used [33,38].

**Image Quality Score:** For qualitative assessment, coronal maximum intensity projection (MIP) images were reconstructed for all the series for side-by-side blinded comparison of sFBI and cFBI without providing source images. A board-certified radiologist (K.N.) with over 25 years of experience performed visual grading of the sharpness of the vessel using a 0 to 4 scale that was more granular compared to similar existing schemes [39,40]: 0 = unacceptable, with severe noise and/or artifacts and impossible to interpret; 1 = poor, with moderate noise and/or artifacts and difficult to interpret; 2 = average, with mild noise and/or artifacts and interpretable; 3 = good, with nearly no noise and/or artifacts and easily interpretable; 4 = excellent. To determine inter-reader agreement, an additional radiologist (Y.K.) with over 7 years of experience in imaging of vascular diseases, performed the same reading. In cases of disagreement (e.g., readers giving different grades for the vessel sharpness), the grade from the senior radiologist was used for the remainder of the study.

**SNR and CNR:** For quantitative assessment, at each station, a diastole source image slice showing the largest cross-section of the major artery was selected. Regions of interest (Figure 2) were placed in the artery, the muscle, and the background to determine the mean and standard deviation (SD) of the signal intensity (SI). The SNR of the artery was determined as the mean SI of the artery divided by the SD of the background. Similarly, the CNR of the artery was determined as the difference in the SNR between the artery and the muscle.

**Statistics:** Effects of acquisition (sFBI vs. cFBI) and station (iliac, femoral, and tibial) on the mean scan time, SNR, CNR, and image quality scores were assessed using two-way repeated measures ANOVA, using JASP (Version 0.18.3) statistics software [41]. In all tests, Mauchly’s sphericity assumption was checked to determine if corrections were necessary. Post hoc comparisons were made using Holm correction for adjusting *p*-values for multiple comparisons. The significance level was set at *α* = 0.05. Inter-reader agreement in grading was assessed using intraclass correlation analysis.

## 3. Results

**Observations:** Visually, there was little difference in the overall quality and contrast between the sFBI (Figure 3A–C) vs. cFBI (Figure 3D–F) coronal MIP images from a healthy volunteer, regardless of the acquisition technique. Between stations, the iliac station tended to exhibit imaging artifacts more than other stations, but this did not seem to compromise the depiction of the iliac artery. While there were minor differences in the depiction of small side-branching arteries between techniques, in both techniques, the major arteries were depicted clearly. The above observations also applied to the PAD patients: the image quality was similar between the sFBI (Figure 4A–C) vs. cFBI (Figure 4D–F) coronal MIP images, and the iliac station exhibited more artifacts. In one PAD patient who has received an iliac stent (Figure 5), it was difficult to acquire good subtraction images using either technique at the iliac station (Figure 5A,D). For the downstream stations, while sFBI exhibited severe Nyquist N/2 aliasing artifacts (Figure 5B,C), cFBI fared much better (Figure 5E,F). CT angiography (Figure 5G) and 3D reconstruction (Figure 5H) demonstrate the presence of the stent.

**Image Quality Score:** Figure 6A shows the image quality score of the arteries seen on coronal MIP images using a 0 to 4 scale. We found a wide distribution of scores from 0 to 4. There was a significant effect of the station (*p* < 0.01) but not the sequence (*p* = 0.8), with a significant interaction (*p* < 0.05): the mean score for the iliac station was lower than that in the femoral station (posthoc *p* < 0.05) and within the iliac station, cFBI had lower mean score than sFBI (posthoc *p* < 0.05). There were no significant differences between femoral vs. tibial (*p* = 1.00) and iliac vs. tibial (*p* = 0.09). The inter-reader agreement in the scores was good, with the intraclass correlation coefficients of 0.790 and 0.739 for sFBI (Figure 6B) and cFBI (Figure 6C), respectively.

**SNR and CNR:** SNR (Figure 7A) and CNR (Figure 7B) values measured on the diastolic images were generally high, averaging from 44 to 133, varying somewhat by the sequence and the station. Between sequences, sFBI had higher SNR (*p* < 0.01) and CNR (*p* < 0.001) values. Iliac station had the lowest SNR and CNR overall: SNR of the iliac station was significantly lower (*p* < 0.01) than that of the tibial station, but no significant differences were found between iliac vs. femoral (*p* = 0.26) or femoral vs. tibial (*p* = 0.38). Similar trends were found for the CNR values—the iliac station showed a trend of lower values compared to tibial (*p* = 0.06), while the iliac vs. femoral (*p* = 0.28) or femoral vs. tibial (*p* = 1.00) stations did not show significant differences. As an exploratory evaluation, SNR and CNR between healthy vs. PAD patients were compared, and we did not find a significant difference (*p* = 0.65 for SNR, *p* = 0.23 for CNR).

**Scan Time (Table 2):** There was a reduction in scan time through the use of cFBI (*p* < 0.001) and when scanning the tibial station vs. the iliac station (post hoc *p <* 0.01). sFBI had average scan times ranging from 136 s to 174 s, depending on the station, while the cFBI had average scan times ranging from 83 to 106 s, roughly a 40% reduction in time. Compared to the scan time for the tibial station, the times for the femoral and iliac stations took on average 16% and 28% longer due to wider coverage in the slice direction, respectively.

## 4. Discussion

This study determined the effect of sFBI vs. cFBI, as well as different anatomic stations, by comparing scan time, image score, SNR, and CNR when imaging the peripheral arteries of the lower extremities in healthy subjects and patients with PAD. We found a marked, nearly 40% reduction in scan time from the use of cFBI without a loss of image quality (by scoring), even with a slight decrease in SNR and CNR. In one PAD patient with a stented aortoiliac artery, we observed in cFBI a marked reduction in N/2 aliasing artifacts at the femoral station and motion artifacts in the tibial station, which were present in the sFBI.

Balancing scan duration with image quality is essential in clinical settings. Although reducing scan time improves patient comfort and workflow efficiency, it must not come at the cost of image quality, SNR, or CNR, as these factors influence diagnostic reliability. As such, protocol optimization to maintain both efficiency and image quality is critical. The overall SNR and CNR values were slightly lower for the cFBI compared to sFBI. This can be attributed to a longer acquisition window per shot of cFBI; while both cFBI and sFBI used similar TR values (depending on the subject’s heart rate) of about 1200 to 2800 ms (average ~1900 ms), cFBI had less time after the acquisition window to recover the longitudinal magnetization compared to sFBI. This can be magnified in some subjects with a higher HR of 75 bpm. Appendix A shows the mean SNR values when grouped by HR of less than vs. greater than 75 bpm. Here, while failing to reach statistical difference (*p* = 0.19), a trend of lower SNR in those in the higher HR group was seen. Although the cFBI permits restoration of longitudinal magnetization by applying exponential refocusing pulses with lowered refocusing flip angles in distal echoes [31,32], the higher HR makes it difficult to recover longitudinal magnetization after the RO acquisition due to the saturation effect of inflowing arterial blood. For instance, HR of 96 bpm gives a repletion time (TR) or 2R-R intervals of 1250 ms. (Appendix A shows the dependence of TR on the HR.) The acquisition window of cFBI was 340 ms (e.g., ETL of 68 × ETS of 5 ms = 340 ms) in the diastolic triggering and resulted in less recovery time or saturating inflowing blood signals. Additionally, note that we observed a noticeable decrease in CNR of cFBI, which may be contributed to by shorter T1 of the background muscle tissue signals compared to longer T1 of blood signals.

However, despite cFBI showing slightly lower SNR and CNR than sFBI, this did not affect how readers scored the overall image quality. This apparent discrepancy can be attributed to the improved subtraction efficiency and reduced artifact burden achieved with cFBI. The centric *ky*-*kz* acquisition prioritizes central k-space early in the cardiac cycle, preserving key contrast information and minimizing the effects of motion and flow variability during subtraction. However, the image quality of the iliac arteries in cFBI was inferior to that of sFBI. This can be due to longer acquisition window of centric *ky*-*kz* trajectory, which cannot depict the faster flow of the iliac arteries. Moreover, the use of asymmetric Fourier imaging and the omission of corner k-space lines in cFBI shorten the overall scan time, although the acquisition window is longer in the zigzag centric *ky*-*kz* trajectory, and reduce the accumulation of phase inconsistencies, particularly in regions affected by high flow or susceptibility artifacts, such as below the aortoiliac stents. As a result, although cFBI exhibits slightly reduced SNR and CNR, it maintains vessel conspicuity and diagnostic utility relative to sFBI. With further work, the current 3-to-4 min sFBI protocol could potentially be replaced by the faster 1-to-2 min cFBI protocol without compromising image quality or diagnostic confidence.

The findings of this study are consistent with prior research underscoring the difficulty of optimizing MRI protocols for vascular imaging. While certain CE-MRA techniques can be acquired rapidly in approximately 30 s or less, without the preparation time of the injection root [42], our cFBI method, though slower, remains competitive. CE-MRA relies heavily on precise bolus timing, requiring consistent and reproducible contrast injection as well as rapid acquisition. Inadequate technique can lead to venous contamination and compromise image quality. In contrast, the FBI approach is not subject to these limitations. When compared to other non-contrast MRA methods, such as velocity-selective MRA (VS-MRA) [20,43], our FBI techniques also performed favorably. VS-MRA for the lower extremities typically involves longer scan times of 5-to-8 min per station and yields a lower artery-to-muscle CNR of approximately 30 [43]. Other popular techniques, such as QISS, also take about 7 min to cover the pelvis to the ankles using ten axial stations, excluding each shimming per station [37]. In our study, the mean scan times for all three stations were less than 8 and 5 min for sFBI and cFBI, respectively. Nevertheless, many non-contrast MR angiography techniques, including QISS and FBI, rely on ECG-triggering delays on cardiac phase dependency. Thus, PAD patients with arrhythmia and/or faster HR over 75 bpm would find it difficult to depict arterial blood images. Moreover, PAD patients with stents have no pulsation in the stent with independent flow; both sFBI and cFBI failed in depicting stented vessels. Further development of a cardiac phase-independent technique is required.

Major strengths of this study include the demonstration of rapid, three-station NC-MRA of the entire lower extremity using fresh blood imaging techniques, which effectively acquires images of the peripheral arteries without the use of contrast agents, important for avoiding nephrogenic systemic fibrosis in those with compromised kidney disease [44], common in diabetic patients [45] who often also have PAD [46]. Our novel cFBI sequence significantly reduces total scan time by 40% compared to the sFBI, improving patient comfort and decreasing the likelihood of motion artifacts while maintaining comparable diagnostic image quality. Additionally, the study provides a detailed quantitative comparison of image quality, SNR, and CNR between the methods, supporting the practical benefits and clinical implementation of cFBI for PAD evaluation.

This preliminary study has several limitations. First, although a few participants had peripheral arterial disease (PAD), the majority were healthy individuals, and the study evaluated only the image quality but not diagnostic accuracy. While both sequences (sFBI and cFBI) successfully depicted peripheral vasculature in healthy subjects, a larger study involving more PAD patients is necessary to determine whether any diagnostic differences exist between the two methods, as well as other reference techniques. This work represents an initial step aimed at refining the sequences in a small sample before conducting broader validation. Second, the sample size was limited (n = 13); a larger cohort is needed to confirm these findings and improve the generalizability of the results. Statistically, non-significant differences in the image quality scores between sequences should be interpreted in light of the small sample size that may fail to detect true differences even when they exist. Third, potential confounding factors such as age and sex were not addressed and should be investigated further in future studies with a larger population. Fourth, while further investigation is needed, we found a trend of decreased SNR in cFBI images of subjects with a higher heart rate > 75 bpm. Given that PAD patients may often have comorbid cardiovascular conditions such as tachycardia, this limitation may restrict the clinical applicability. Fifth, we did not evaluate intra-reader agreement; however, in most cases, inter-reader agreement is worse than intra-reader agreement when evaluating images. Lastly, image evaluation relied solely on the senior reader’s grading when there were discordant grades, without joint re-evaluation. This may introduce individual bias that may have increased subjective variability. Fortunately, there were not a large number of discordant cases, evident by good inter-reader agreement (Figure 6).

## 5. Conclusions

In summary, this study confirmed the feasibility of FBI-based methods for imaging peripheral arteries across the three lower extremity stations and showed that the cFBI technique offers a substantial time savings of approximately 40% without compromising image quality. The shorter scan time enhances patient comfort and decreases the likelihood of motion artifacts in clinical practice. Future research involving larger cohorts of PAD patients will be necessary to validate and broaden the applicability of these findings. Ultimately, a clear understanding of how different imaging techniques impact image quality, diagnostic accuracy, and workflow efficiency will enable clinicians to customize protocols for optimal patient care.

## Figures and Tables

**Figure 1 sensors-25-07429-f001:**
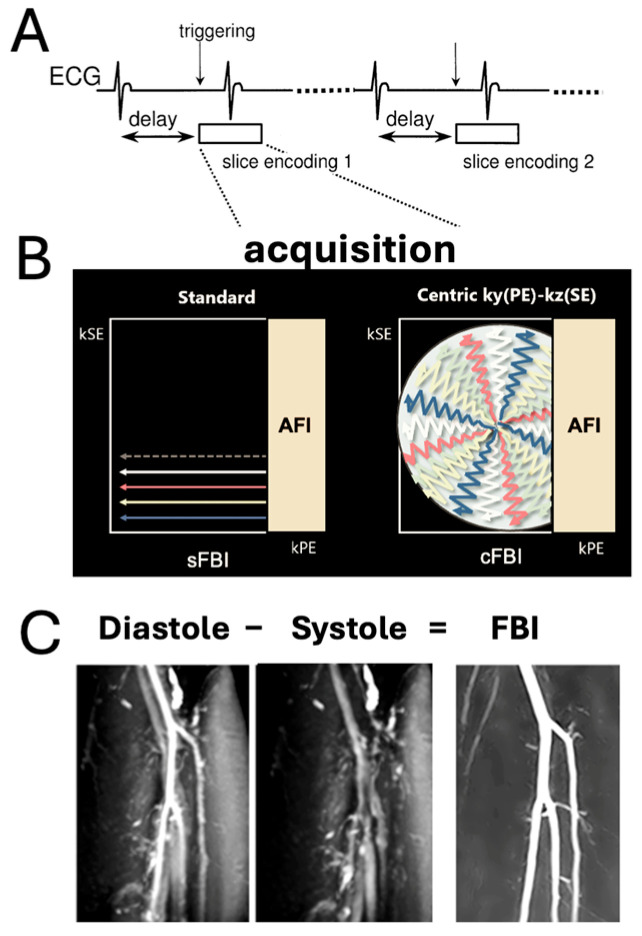
(**A**) Sequence diagram of electrocardiogram (ECG)-synchronized 3D single shot fast spin echo (SSFSE) with a Cartesian readout in the frequency-encoding (readout) direction (through plane). (**B**) Standard fresh blood imaging FBI (sFBI): The sequence is ECG-triggered for each slice encoding to ensure acquisition of all partitions at the same cardiac phase, enabling 3D volume reconstruction. (**B**) Centric *ky*-*kz* FBI (cFBI): The sequence utilizes a zigzag centric *ky*-*kz* trajectory to fill the 3D k-space. Both acquisitions incorporate asymmetric Fourier imaging (AFI) in the phase-encoding direction. Additionally, the cFBI method trims the outer corners of k-space with zero fillings. (**C**) FBI images are generated by subtracting systolic-triggered images from diastolic-triggered images to enhance vascular contrast and suppress background tissue signals.

**Figure 2 sensors-25-07429-f002:**
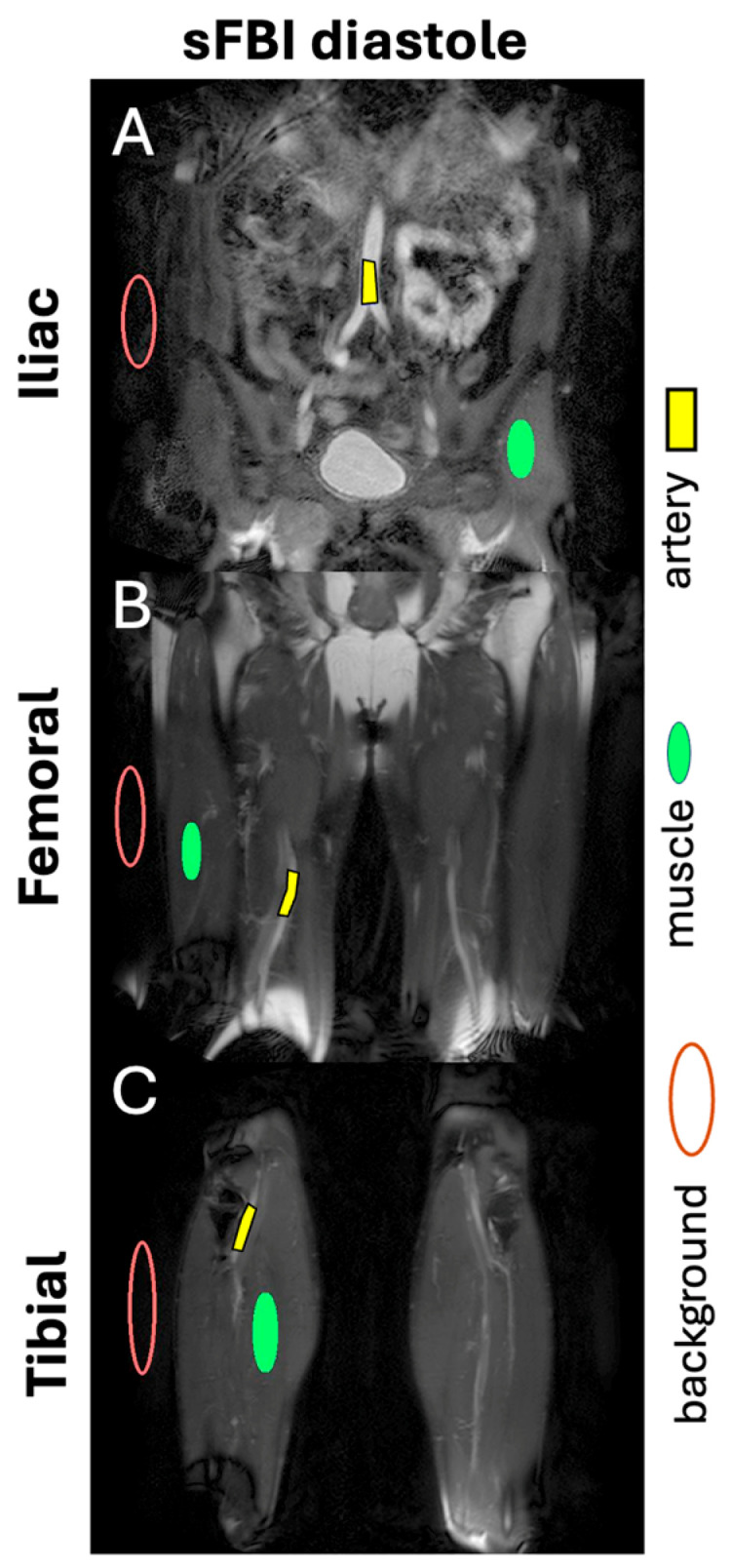
Example of regions of interest for the artery, muscle, and background, placed on diastole images of (**A**) iliac, (**B**) femoral, and (**C**) tibial stations to determine signal-to-noise ratio and contrast-to-noise ratio.

**Figure 3 sensors-25-07429-f003:**
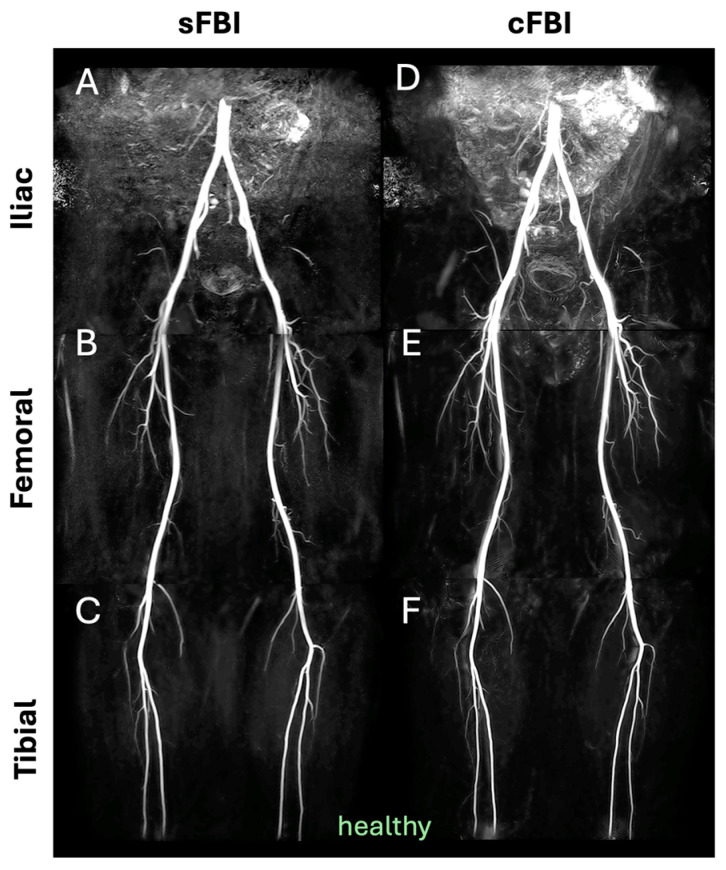
Coronal maximum intensity projection (MIP) images from a healthy volunteer (subject #6, 32-year-old male), acquired with standard (**A**–**C**) and centric (**D**–**F**) fresh blood imaging techniques, at the three stations of iliac (**A**,**D**), femoral (**B**,**E**), and tibial (**C**,**F**).

**Figure 4 sensors-25-07429-f004:**
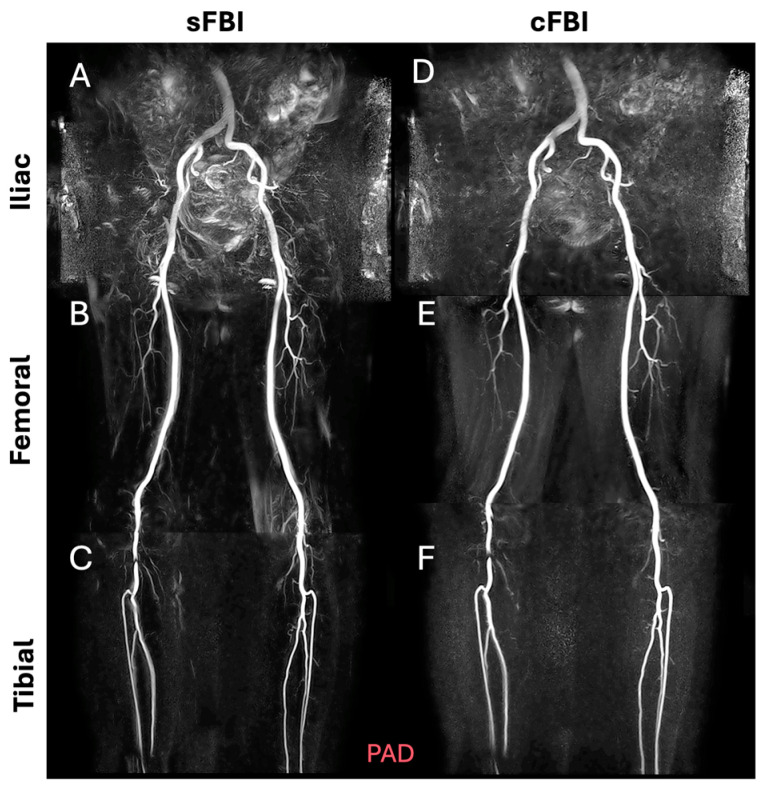
Coronal maximum intensity projection (MIP) images from a patient with peripheral artery disease (subject #13, 74-year-old male), acquired with standard (**A**–**C**) and centric (**D**–**F**) fresh blood imaging techniques, at the iliac (**A**,**D**), femoral (**B**,**E**), and tibial (**C**,**F**) stations. Both tibial stations (**C**,**F**), showing a stenotic appearance, were confirmed as banding artifacts from the source images, which may be due to the metallic properties of the outerwear.

**Figure 5 sensors-25-07429-f005:**
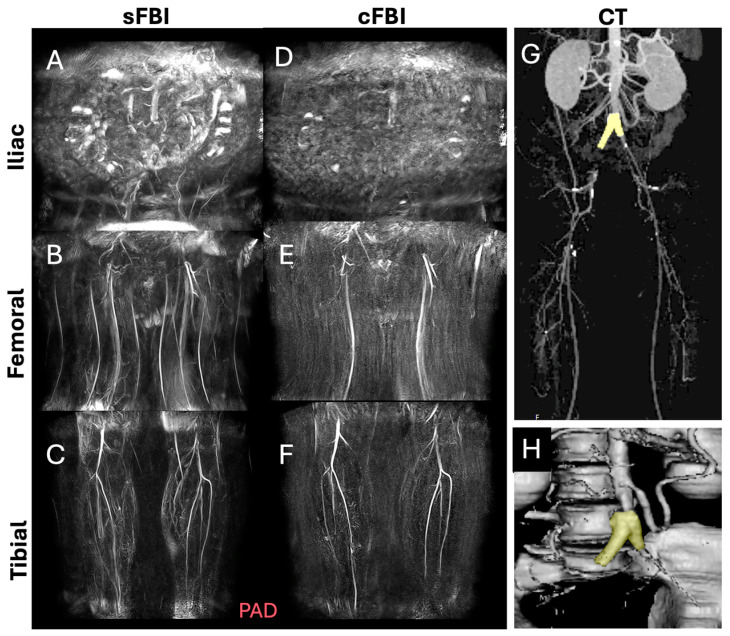
Coronal maximum intensity projection (MIP) images from a patient with an iliac artery treated with a stent (subject #11, 46-year-old male), acquired with standard fresh blood imaging or sFBI (**A**–**C**) and centric *ky*-*kz* fresh blood imaging or cFBI (**D**–**F**) techniques, at the iliac (**A**,**D**), femoral (**B**,**E**), and tibial (**C**,**F**) stations. Corresponding to a computed tomography angiography (**G**) and a 3D reconstruction (**H**) showing the stent (yellow) at the aorto-iliac artery bifurcation. Note the Nyquist (N/2) aliasing artifacts in the lower stations of sFBI (**B**), but not in cFBI (**E**). High-velocity flow jets through the stent can introduce phase shifts that, when not properly flow-compensated in the readout direction, appear as aliasing artifacts. sFBI is more susceptible to this due to its standard k-space acquisition, which spreads high flow-related phase errors in the phase encode direction. In contrast, cFBI uses a zigzag centric *ky*-*kz* trajectory that prioritizes acquisition of central k-space immediately after gating, reducing the impact of flow-related distortions on image contrast. In addition, sFBI in (**C**) shows motion artifacts as the background signals remained from systolic and diastolic subtraction images, while cFBI in (**F**) shows no motion artifacts.

**Figure 6 sensors-25-07429-f006:**
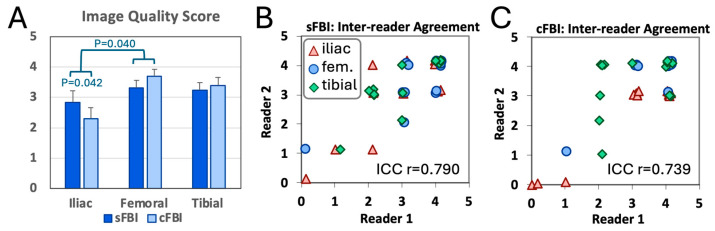
(**A**) Image quality score of the arteries evaluated on coronal maximum intensity projection images by a senior radiologist. Mean +/− standard deviation, n = 13 subjects. Inter-reader agreement between two radiologists for the images acquired with (**B**) standard and (**C**) centric *ky*-*kz* fresh blood imaging.

**Figure 7 sensors-25-07429-f007:**
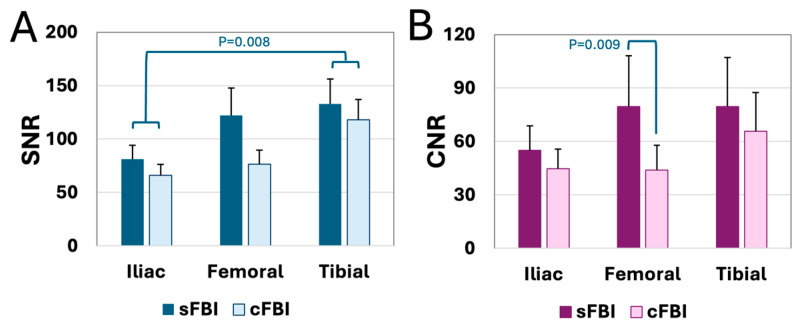
(**A**) Signal-to-noise ratio (SNR) and (**B**) contrast-to-noise ratio (CNR) of the arteries at each station acquired with standard and centric *ky*-*kz* fresh blood imaging sequences. Mean + standard deviation, *n* = 13 subjects.

**Table 1 sensors-25-07429-t001:** Subject characteristics. S.D. = standard deviation. PAD = peripheral artery disease.

Subject ID	Age (years)	Sex	HR (bpm)	BMI (kg/m^2^)
1	26	F	55	18.6
2	23	F	50	29.3
3	51	M	64	24.5
4	32	M	90	25.2
5	39	F	57	20.2
6	32	M	64	25.4
7	52	M	82	25.1
8	79	F	78	25.7
9	50	F	61	23.2
10	58	M	66	23.7
11 (PAD)	46	M	96	30.2
12 (PAD)	53	M	77	23
13 (PAD)	74	F	44	21.7
mean	47.3	6 female	68.0	24.3
S.D.	17.1	7 male	15.6	3.2

**Table 2 sensors-25-07429-t002:** Scan time of each technique at each station and the combined three-station total in seconds. Mean (and standard deviation). *p*-value for the sequence (standard vs. centric *ky*-*kz* fresh blood imaging sequences) was <0.001, the station (iliac, femoral, calf) was 0.004, and the interaction term was 0.66.

	sFBI [s]	cFBI [s]
Iliac	173.8 (34.1)	105.9 (30.4)
Femoral	158.4 (38.3)	95.5 (30.0)
Calf	136.2 (31.0)	82.7 (22.3)
3 Station Total	468 (88)	291 (80)

## Data Availability

The data that support the findings of this study are not publicly available due to reasons of sensitivity. Anonymized data may be available from the corresponding author upon review of the request. Data are located in controlled-access data storage at the corresponding author’s institution.

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
