# Peer review of "Three-Station Non-Contrast MR Angiography of the Lower Extremities Using Standard and Centric Fresh Blood Imaging"

_sensors, 2025, doi:10.3390/s25247429_

Round 1
Reviewer 1 Report
Comments and Suggestions for Authors
This study optimized the technology for non-contrast MR angiography (NC-MRA) in lower extremity arterial disease (PAD), confirming that the new central ordering technology (cFBI) can shorten the scanning time while maintaining diagnostic quality compared with the traditional method (sFBI). It directly addresses the two major clinical pain points: contraindications to the use of contrast agents and long scanning time, with a clear research design. However, the study has obvious shortcomings in terms of sample size, verification of diagnostic efficacy, and applicability to special populations. More data need to be supplemented to enhance the reliability of the conclusions and the value of clinical promotion. In summary, our specific comments are as follows:
- The total sample size was only 13 cases (10 healthy volunteers + 3 PAD patients), with only 1 PAD patient having an iliac artery stent. The sample of only 3 PAD patients is insufficient to support the conclusion that "cFBI is suitable for the PAD population."
- The study did not use Digital Subtraction Angiography (DSA) or CTA as a gold standard to validate the diagnostic sensitivity, specificity, positive predictive value (PPV), and negative predictive value (NPV) of cFBI for PAD lesions (such as stenosis, occlusion, and collateral circulation). Therefore, it is impossible to confirm whether its "diagnostic accuracy" is comparable to existing clinical techniques.
- The paper mentions a "trend toward decreased SNR for cFBI in patients with high heart rates (>75 bpm) (p=0.19)," but no subgroup analysis was performed (e.g., categorizing patients into <60 bpm, 60-75 bpm, and >75 bpm groups to quantify the impact of heart rate on SNR/CNR). Given that PAD patients often have comorbid cardiovascular conditions (e.g., atrial fibrillation, tachycardia), this limitation significantly restricts the clinical applicability of cFBI.
- The paper states that "in case of disagreement, the rating of the senior physician was adopted," but it does not define what constitutes a "disagreement." Relying solely on a senior physician's decision, without having two or more physicians jointly re-evaluate the discordant images, may introduce individual bias and lacks a process to minimize subjective variability.
Reviewer 2 Report
Comments and Suggestions for Authors
Comments to Sensors-3981574
Entitled “Three Station Non-Contrast MR Angiography of the Lower Extremities Using Standard and Centric Fresh Blood Imaging”
General Comments:
The authors have constructed comprehensive research, but there is still a need for revision.
Specific Comments:
- In the abstract:
- In the background:
What is the primary aim of this study?
- In the Materials and Methods:
- In Lines 21-23: “in 10 healthy subjects and three patients with PAD (age range: 23–79 years; 7 females) using a 3-Tesla MRI system. So, how many females, including healthy and PAD patients, were in your study?
- In Table 1, what is the BMI value for subjects 5 to 13?
- In Figure 2, are there any differences in the regions of interest (ROI) for the muscle and background between the femoral and tibial? Why did you select the tibial area as background, and not the femoral?
- So, continue as above. Why do you need to have three stations to compare sFBI and cFBI? Why not iliac and femoral or iliac and tibial?
- In Figure 6, why is the iliac sFBI higher than the cFBI in the image quality score, but the femoral and tibial sFBI are lower than the cFBI? Is the difference between the sFBI and cFBI statistically significant? If yes, please add the p-value in the figure. How about the intra-reader agreement in sFBI and cFBI?
- In Figure 7(A) and (B), please add the p-value for the iliac, femoral, and tibial.
- In the discussion:
- What are the advantages and disadvantages of 2D and
3D (times of flight) TOF MRA in access PAD?
(2) Please add one paragraph before the limitations of this study and illustrate the strength of your research.
Reviewer 3 Report
Comments and Suggestions for Authors
This manuscript presents a comparative study of standard (sFBI) and centric (cFBI) fresh blood imaging techniques for three-station non-contrast MR angiography of the lower extremities.The study is well-motivated, timely, and addresses an important clinical need for safer and faster vascular imaging.The following issues need improvement:
1.The use of repeated-measures ANOVA is appropriate. However, more details on post-hoc tests and correction for multiple comparisons would be helpful. Also, the non-significant difference in image quality scores between sequences should be discussed in the context of the small sample size.
2.The clarity and scale of some images need to be adjusted, such as Figure 6.
3. Some acronyms () are not fully defined on first use. Some are defined in the captions of the images, but not in the main text.
Round 2
Reviewer 1 Report
Comments and Suggestions for Authors
The authors have made revisions and additions to the issues I was concerned about, and the paper can be accepted.
Author Response
Thank you for positive comments!
Reviewer 2 Report
Comments and Suggestions for Authors
Comments to Sensors-3981574 (V2)
Entitled “Three Station Non-Contrast MR Angiography of the Lower Extremities Using Standard and Centric Fresh Blood Imaging”
General Comments:
The authors have conducted comprehensive research, but there is still room for revision.
Specific Comments:
- In the Materials and Methods:
- In Figure 6 (A), please add all the p-values in the figure, that is, just as p=0.042 in Iliac, how about the p-value in Femoral and Tibial? You add p=0.040 between the Iliac and Femoral, but how about between the Femoral and Tibial? And how about between the Iliac and the Tibia?
- You only list the inter-reader agreement, but how about the intra-reader agreement in sFBI and cFBI? If you cannot list, please explain the reason.
- In Lines 263, the intraclass correlation coefficients should be revised as “the interclass correlation coefficients.”
- In Figure 7(A) and (B),
In SNR, the relationship between sFBI and cFBI is similar
to that in CNR, whether in iliac, femoral, or tibial. The p-
values for comparing iliac, femoral, or tibial between sFBI
and cFBI, or between iliac and femoral, femoral and tibial,
or iliac and tibial, should be presented entirely in a Figure or a textual description. But it is better presented in the figure.
